# Use of standardised patients to assess tuberculosis case management by private pharmacies in Patna, India: A repeat cross-sectional study

Anita Svadzian[1,2], Benjamin Daniels[3], Giorgia Sulis[4], Jishnu Das[3,5], Amrita Daftary[6,7], Ada Kwan[8], Veena Das[9], Ranendra Das[10], Madhukar Pai[1,2,11] *

1 Department of Epidemiology, Biostatistics and Occupational Health, McGill University, Montreal, Quebec, Canada, 2 McGill International TB Centre, McGill University, Montreal, Quebec, Canada, 3 Georgetown University, Washington, DC, United States of America, 4 School of Epidemiology and Public Health, Faculty of Medicine, University of Ottawa, Ottawa, Ontario, Canada, 5 Centre for Policy Research, New Delhi, India, 6 Dahdaleh Institute of Global Health Research, School of Global Health, York University, Toronto, Ontario, Canada, 7 Centre for the Aids Programme of Research in South Africa MRC-HIV-TB Pathogenesis and Treatment Research Unit, Durban, South Africa, 8 Division of Pulmonary and Critical Care Medicine, Department of Medicine, University of California, San Francisco, San Francisco, California, United States of America, 9 Department of Anthropology, Johns Hopkins University, Baltimore, Maryland, United States of America, 10 Institute for Socio-Economic Research on Development and Democracy, Delhi, India, 11 Manipal McGill Program for Infectious Diseases, Manipal Centre for Infectious Diseases, Manipal Academy of Higher Education, Manipal, Karnataka, India

* madhukar.pai@mcgill.ca

**Data Availability Statement:** All study instruments, tools, and datasets from individual SP studies are freely accessible from the QuTUB

## Abstract

As the first point of care for many healthcare seekers, private pharmacies play an important role in tuberculosis (TB) care. However, previous studies in India have showed that private pharmacies commonly dispense symptomatic treatments and broad-spectrum antibiotics over-the-counter (OTC), rather than referring patients for TB testing. Such inappropriate management by pharmacies can delay TB diagnosis. We assessed medical advice and OTC drug dispensing practices of pharmacists for standardized patients presenting with classic symptoms of pulmonary TB (case 1) and for those with sputum smear positive pulmonary TB (case 2), and examined how practices have changed over time in an urban Indian site. We examined how and whether private pharmacies improved practices for TB in 2019 compared to a baseline study conducted in 2015 in the city of Patna, using the same survey sampling techniques and study staff. The proportion of patient-pharmacist interactions that resulted in correct or ideal management, as well as the proportion of interactions resulting in antibiotic, quinolone, and corticosteroid are presented, with standard errors clustered at the provider level. To assess the difference in case management and the use of drugs across the two cases by round, a difference in difference (DiD) model was employed. A total of 936 SP interactions were completed over both rounds of survey. Our results indicate that across both rounds of data collection, 331 of 936 (35%; 95% CI: 32–38%) of interactions were correctly managed. At baseline, 215 of 500 (43%; 95% CI: 39–47%) of interactions were correctly managed whereas 116 of 436 (27%; 95% CI: 23–31%) were correctly managed in the second round of data collection. Ideal management, where in addition

consortium (https://www.qutubproject.org/), funded by the Bill & Melinda Gates Foundation.

**Funding:** This study was funded by the Bill & Melinda Gates Foundation (grant OPP1091843), and the Knowledge for Change Program at the World Bank. The funders had no role in the manuscript preparation, analysis, or submission.

to a referral, patients were not prescribed any potentially harmful medications, was seen in 275 of 936 (29%; 95% CI: 27–32%) of interactions overall, with 194 of 500 (39%; 95% CI: 35–43%) of interactions at baseline and 81 of 436 (19%; 95% CI: 15–22%) in round 2. No private pharmacy dispensed anti-TB medications without a prescription. On average, the difference in correct case management between case 1 vs. case 2 dropped by 20 percent points from baseline to the second round of data collection. Similarly, ideal case management decreased by 26 percentage points between rounds. This is in contrast with the dispensation of medicines, which had the opposite effect between rounds; the difference in dispensation of quinolones between case 1 and case 2 increased by 14 percentage points, as did corticosteroids by 9 percentage points, antibiotics by 25 percentage points and medicines generally by 30 percentage points. Our standardised patient study provides valuable insights into how private pharmacies in an Indian city changed their management of patients with TB symptoms or with confirmed TB over a 5-year period. We saw that overall, private pharmacy performance has weakened over time. However, no OTC dispensation of anti-TB medications occurred in either survey round. As the first point of contact for many care seekers, continued and sustained efforts to engage with Indian private pharmacies should be prioritized.

## Introduction

Tuberculosis (TB) is a leading infectious cause of death globally, with cases projected to rise even more due to the disruptions caused by the COVID-19 pandemic [1]. In 2021, only 6.4 of 10.6 million people with TB were detected and notified to national TB programs [2]. These "missing" patients, individuals who are either undiagnosed or not notified to the TB programme, are estimated to amount to 4.2 million cases yearly. Failing to identify these persons results in continued community transmission, undermining national and global TB control efforts [3].

India has the world's highest burden of tuberculosis (TB) and "missing patients" globally [2]. In high-incidence settings, it has been shown that patients with TB frequently experience delays before obtaining a diagnosis [4]. On average, a patient with TB visits three providers, experiencing a delay of 55 days before being diagnosed and initiating TB treatment [4, 5]. With over 750,000 retail outlets in India, private pharmacies could play an important role in case detection and referral with the potential to plug early leaks in the TB care cascade and reducing transmission and incidence, by strengthening the steps between symptom screening, diagnosis and case notification [6, 7]. In India, private pharmacies are an attractive access point for medical services with their long operating hours, drug inventory, lack of queues and consultation fees [8, 9]. Given their ubiquity, accessibility and trusted role in local communities [10], private pharmacies are frequently the first point of contact for patients soon after they develop symptoms of TB [11–13]. In this context, up to 40% of patients who were ultimately diagnosed with TB visited a private pharmacy prior to medical assessment and TB diagnosis, with 25% of patients continuing to seek advice from private pharmacies even after diagnosis, preferring to avoid doctor consultations [14–16].

While private private pharmacies are positioned to play an important role in case detection and referral, inappropriate management often contributes to delayed diagnosis and TB-specific treatment [14]. Previous studies in India have shown that private pharmacies commonly

dispense cough syrups, bronchodilators, anti-histamines and antibiotics over-the-counter (OTC), rather than referring patients to the appropriate provider for TB testing and treatment [17–19]. Such practices of self-medication and poor referral practices can delay TB diagnosis [4, 5]. Private pharmacies that dispense inappropriate antibiotics to patients with TB may also contribute to the development of antimicrobial resistance (AMR). Private private pharmacies have thus been identified as key stakeholders in both TB and AMR control efforts [12].

In a standardized patient study by Satyanarayana et al. from our group conducted in 2015, it was found that while no private pharmacy dispensed anti-TB medications without a prescription, 319 (27%) of 1,200 (95% CI: 24–29%) patient-pharmacist interactions resulted in broad-spectrum antibiotic dispensing, across all interactions [18]. The results also suggested that the use and misuse of antibiotics are influenced by drug category and the information that care seekers present. Findings also showed that 38% of the private pharmacies dispensed antibiotics or corticosteroids to people with TB symptoms but no test results. In addition, there was dispensation of quinolones in 7% and corticosteroids in 5% of interactions.

Since this baseline study, there have been numerous attempts to engage pharmacists in TB care, treatment education, and screening and referral [1, 19–23], and while this study was not aimed to parse out the individual impact of each respective intervention, we did expect them to have changed practice amongst pharmacists in these settings to some extent.

In this current study, we again made use of standardized patients (SPs), a healthy individual trained to portray a clinical problem or situation for the purpose of better understanding specific behaviours. We assessed over-the-counter medical advice and drug dispensing practices of pharmacists for standardized patients presenting with classic symptoms of pulmonary TB and for those with sputum smear positive pulmonary TB disease, and examined how practices have changed over time in Patna, surveying the majority of the same provider (92% same private pharmacies). Specifically, we assessed how and whether private pharmacies are improving practices for suspected TB compared to a baseline study conducted in 2015 compared to the same provider behaviours in 2019 [18].

## Context

This study was conducted in Patna, the capital of the state of Bihar. With a population of approximately 2 million urban dwellers and a per capita income of 30,441 INR (US$412), it remains one of India's least developed urban centers. Overall, 106,189 patients with TB were notified in Bihar [1] and within the public sector, 43,139 (40.6%) cases were confirmed via microbiological testing. The National TB Prevalence Survey (2019–21) reported a TB prevalence of 327 per 100,000 in Bihar state, and also showed that there were many missing TB cases in 2021; it was found that 3.15 TB cases were missed per 1 case notified to the public system [24].

## Materials and methods

### Study design

We assessed the medical advice and drug dispensing practices of private pharmacies for standardised patients presenting with either presumptive TB (case 1) or microbiologically confirmed tuberculosis (case 2). By assessing the difference in antibiotic use across the two cases for the same pharmacists, we broke down the relative importance of antibiotic misuse arising from the lack of diagnosis (case 1) versus antibiotic use despite a confirmed diagnosis for which antibiotics are contraindicated (case 2).

Standardized patients (SPs) were used in this study to assess the quality of TB care. SP methodology is being used more and more in low-resource settings due to its advantages over

alternative methods. Alternative data collection methods, such as questionnaire surveys of knowledge, patient surveys, and chart analyses etc., rely on participant recall and are thus prone to inherent biases [25–27]. In contrast, SP studies are less vulnerable to recall bias, and have been shown to provide accurate assessments of provider practice without observation bias [28–30].

To set the benchmark for what pharmacists should do when faced with such patients, the guidelines from the Government of India's TB Control program and the Indian Pharmaceutical Association were used [31]. These guidelines specify that private pharmacies should counsel patients about tuberculosis, identify and refer persons with tuberculosis symptoms to the nearest public health facilities for tuberculosis testing, and play a part in the provision of tuberculosis treatment. Therefore, pharmacists adhering to these guidelines should have referred the standardised patients to healthcare providers without dispensing either antibiotics or corticosteroids, both of which require a prescription.

Recruitment of SPs, script development, SP training, provider sampling and assignment of SP case providers was previously outlined by Satyanarayana [18] and will thus not be re-described here.

## Standardised patient presentations

Case presentations are summarized in Table 1. Standardised patients trained as case 1 presented to pharmacists with 2–3 weeks of cough and fever and sought relief directly from the private pharmacy (i.e. seeking medicines). This case presentation could be indicative of TB or other chronic respiratory infections. While an antibiotic might be warranted for some of these conditions, giving one to the SP would not be correct management without a prescription from a doctor, rather, a referral for TB testing would be.

Standardised patients trained as case 2 presented with 1 month of cough and fever and a tuberculosis positive laboratory report from a recent sputum smear test from a government healthcare provider. While tuberculosis was indeed confirmed by the laboratory report, the standardised patients would present as a naïve patient, and make it clear that they did not fully understand what the report said. The pharmacist should in this situation recognize the futility of broad-spectrum, short-term antibiotic dispensation but could still offer them (e.g. if driven by profits) since the presenting SP has made their confusion with their diagnosis of TB clear.

Neither of the standardised patient cases presented with drug prescriptions. After each private pharmacy visit, standardised patients were debriefed with a structured questionnaire within 1 hour of the visit.

## SP training

The training of SPs ensured that they (a) correctly presented the cases, (b) correctly recalled the interaction with the private pharmacy staff, and (c) avoided detection. The first two aims were achieved through classroom training in case presentation and testing of recall, as well as mock interviews and dry runs that were supervised in the field. For the third aim, SPs were taught to avoid detection by the following methods. First, our recruitment strategy ensured that SPs came from low-income areas or slums from the same cities in which the project was located, and the areas from which they came were far from the field sites. This meant that their clothing, mannerisms, and speech were very close to the ordinary patients who visited pharmacists, but they would not have been personally known in the study areas. Second, previous observations in private pharmacies and pharmacist shops were conducted by supervisors in order to observe the patterns of interaction (e.g., mode of address), and we ensured that SPs approximated those patterns of interaction. Third, during the training, SPs were taught to

**Table 1. Standardised patient case descriptions.**

| | Case description | Presentation of standardised patient | Expected case management |
|---|---|---|---|
| Case 1 | A possible case of tuberculosis, presenting with 2–3 weeks of cough and fever and directly seeking care from a pharmacist | Case 1 presents with the opening statement, "Sir, I have cough and fever that is not getting better. Please give me some medicine." At presentation, this case has had a 2–3 week cough, which occurred more during early morning and night, accompanied by a 2–3 week, on-and-off, low-grade fever. The patient was producing sputum that did not contain any blood. The case would admit to a loss of appetite and to his or her clothes becoming a bit loose if prompted by the pharmacist. If the pharmacist asked about taking medicines for this illness, the patient would say no | Verbal or written referral to a DOTS centre or a health-care provider without dispensing any antibiotics (including anti-tuberculosis drugs and quinolones) or corticosteroids |
| Case 2 | Chronic cough with a positive sputum smear report for tuberculosis from a government dispensary and directly seeking care from a pharmacist | Case 2 presents with a positive sputum smear result visiting a pharmacist, presenting with the opening statement, "Sir, I am having cough for nearly a month now and also have fever." While showing a positive sputum report to the pharmacist, the patient continues, "I went to the government dispensary and they asked me to get my sputum tested. I have this report. Can you please give me some medicine?" At presentation, this case has had a cough for 1 month and produces sputum without blood, accompanied by a 1 month, on-and-off, low-grade fever, which was more during evening times. Similar to Case 1, the case would admit to a loss of appetite and to his or her clothes becoming a bit loose if prompted by the pharmacist. If the pharmacist asked about taking medicines for this illness, the patient would say no | Verbal or written referral to a DOTS centre or a health-care provider without dispensing any antibiotics (including anti-tuberculosis drugs and quinolones) or corticosteroids |

DOTS = directly observed treatment, short-course.

internalize completely the characters and the details of their mock stories through which the character was made alive to them. In mock interviews during training, supervisors would add unscripted questions with regard to family or neighborhood that SPs could answer spontaneously because they were of the actual social background that was being approximated in the characters they were portraying. Finally, dry runs were conducted in which the supervisor was present in the shop on the pretense of buying something such as toothpaste or an over-the-counter cough syrup and thus could watch the interaction and offer corrections later.

## Selection of private pharmacies, standardised patient visits and study size

In the 2014 round, SPs completed 500 visits to private pharmacies in Patna. In the repeat survey, standardized patients were sent to 260 randomly sampled private pharmacies in Patna between October 2019 and January 2020, resulting an additional 436 interactions (936 total interactions between baseline and second survey). This sample size was large enough to allow a precise estimate of the outcome of interest using the Clopper-Pearson exact method (i.e. proportion of visits resulting in ideal management, assumed to be 10% of case 1 interactions 60% of case 2 interactions based on previous study)[15] within a confidence interval of 8.30–11.70% and 57.16–62.79%, respectively. If it is assumed that the various Private-Provider Interface Agencies (PPIAs) initiatives were effective and resulted in a 10% increase in correct management, this sample size would allow an estimate of the prevalence difference between 5.51–14.49% for case 1 and 4.33–15.67% for case 2.

In urban Patna (defined as Patna, Danapur, and Phulwarisharif blocks), a lane-by-lane mapping exercise conducted between January and August 2014 served as a complete list of

private pharmacies that were operating in these areas at the time. Additionally, urban TB programs implemented by PPIAs were recruiting and enrolling pharmacists or pharmacist assistants into TB referral and treatment networks in Patna. The geographical frame covered all 40 wards in Danapur block, all 28 wards in Phulwari Shariff block, and 34 wards selected in collaboration with the PPIAs out of 73 wards in Patna block. For both of the random samples in Patna, we provided a reserve list, which could replace originally sampled pharmacists found to be permanently closed at the time of data collection for the purposes of surveillance. The same catchment area and mapping used in the baseline study (2014) was used in the survey in 2019; 92% of the providers visited at baseline were revisited in the 2019 survey.

## Identification of drugs given by pharmacists

Guidelines for private pharmacies are specified under the Ministry of Health and Family Welfare's Drugs and Cosmetics Rules Act, 1945 [32]. Schedule H and Schedule H1 contain all antibiotics and corticosteroids. Schedule H drugs cannot be given to patients without a prescription from a qualified medical practitioner. In 2013, regulations were further tightened, with anti-tuberculosis drugs (isoniazid, rifampicin, ethambutol, and pyrazinamide) and some quinolones (such as moxifloxacin and levofloxacin, used in the treatment of tuberculosis) listed on a newly created Schedule H1. For H1 drugs, private pharmacies require both a prescription from a qualified medical practitioner and a separate register to record the name and address of the prescriber, the patient, the names of the drugs and the quantity supplied [33, 34].

All labelled medicines prescribed by private pharmacies were digitized and stored. They were then coded by two doctors with expertise in TB (SS) and infectious diseases (RS). In the repeat survey, coding was replicated by staff who had worked on the previous survey. Blinded from provider identifying details, they identified and categorized medicines as corticosteroids, anti-TB drugs, quinolones, or other broad-spectrum antibiotics under maker-checker procedure, whereby dual-approval was needed by two separate coders for each medicine reviewed. Loose or unlabeled pills were discarded.

## Data sources

All study instruments, tools, and datasets from individual SP studies are freely accessible from the QuTUB consortium (https://www.qutubproject.org/), funded by the Bill & Melinda Gates Foundation.

## Definition of main outcomes

We operationalized our two main, binary outcomes in one of two ways. First, we defined correct case management as verbal or written referral to a DOTS centre or to a healthcare provider, regardless of medicines dispensed. We then defined ideal case management for both cases as a verbal or written referral to a DOTS centre or to a healthcare provider without dispensing any antibiotics (including anti-tuberculosis drugs and quinolones) or corticosteroids.

## Statistical analysis

The proportion of interactions that resulted in correct or ideal management, as well as the proportion of interactions resulting in antibiotic, quinolone, and corticosteroid, is presented. The unit of analysis is a private pharmacy-SP interaction irrespective of who (private pharmacy owners, pharmacists, or private pharmacy assistants) the standardised patient interacted with.

Whether the case was correctly managed was assessed from a tuberculosis perspective, consistent with Standards for Tuberculosis Care in India and International Standards for Tuberculosis Care. We employed ordinary least squares regression to assess differences in outcomes by case.

To assess the difference in case management and the use of drugs across the two cases by round, a difference in difference (DiD) model was employed. DiD is a non-experimental statistical technique used to estimate treatment effects by comparing the change (difference) in the differences in observed outcomes between treatment and control groups, across pre-treatment and post-treatment periods [35].

It is commonly used to recover the causal effect of interest from observational study data—where the experimental design is out of the researcher's control and usually subjected to unobserved confounders and some form of selection bias. DiD is a combination of time-series difference–comparing outcomes across pre-treatment and post-treatment periods–and cross-sectional difference–comparing outcomes between treatment and control groups [35]. In this case, the treatment effect can be estimated by subtracting the average change in the control group from the average change in the treatment group.

With repeated, cross-sectional data gathered in our surveys, the regression model can be defined as:

$$y_{it} = \beta_0 + \beta_1 P_t + \beta_2 T_i + \beta_3 (P_t * T_i) + u_{it}$$

where $y$ is the outcome of interest (ideal or correct case management for instance), $P$ is a dummy variable for the second round of data collection in our survey and $T$ is a dummy variable for the treatment group–i.e. cases 1 or 2. The interaction term, $P \times T$, is equivalent to a dummy variable equal to 1 for observations of case 2 and in the second round of data collection.

The coefficients can be interpreted as follows:

- $\beta_1$: Average change in the outcome from the first to the second survey round that is common to both cases

- $\beta_2$: Average difference in outcome between the two cases that is common in both rounds of the survey

- $\beta_3$: Average differential change in the outcome from the first to the second survey round of case 2 relative to case 1

  All estimates clustered standard errors at the provider level.

## Ethics

This study was granted ethical approval by the McGill University Health Centre in Montreal, Canada (REB No. 14-137-BMB) and the Subcommittee for the Ethical Approval of Projects at the Institute for Socioeconomic Research on Development and Democracy in Delhi, India [17]. A waiver from getting informed consent from pharmacists in Patna was waived under the Government of Canada Panel on Research Ethics, as well as a recent study by Rhodes and colleagues (2012) on the ethical aspects of SP studies appointed by the US Department of Health and Human Services [36]. Further study specific rational for the waiver is detailed elsewhere [37]. Standardized patients were all hired as study staff and explicitly coached to protect themselves from harmful medical interventions. The funders had no role in study design, data collection, data analysis, data interpretation, or writing of the report.

**Table 2. Summary statistics of main outcomes.**

| | Full sample | Baseline (2014–2015) | Round 2 (2018–2019) |
|---|---|---|---|
| *Correct Case Management* | 0.35 | 0.43 | 0.27 |
| | 331 | 215 | 116 |
| | (0.478) | (0.496) | (0.442) |
| *Ideal Case Management* | 0.29 | 0.39 | 0.19 |
| | 275 | 194 | 81 |
| | (0.456) | (0.488) | (0.389) |
| *N* | *936* | *500* | *436* |

Mean coefficients; counts; sd in parentheses; correct case management = verbal or written referral to a DOTS centre or a healthcare provide; ideal case management = verbal or written referral to a DOTS centre or a healthcare provider without dispensing any antibiotics (including anti-tuberculosis drugs and quinolones) or corticosteroids

## Results

Our results, detailed in Table 2, indicate that across both rounds of data collection, 331 of 936 (35%; 95% CI: 32–38%) of interactions were correctly managed; at baseline, 215 of 500 (43%; 95% CI: 39–47%) of interactions were correctly managed whereas 116 of 436 (27%; 95% CI: 23–31%) were correctly managed in the second round of data collection.

Ideal management, where in additional to a referral, patients were not prescribed any potentially harmful medications, was seen in 275 of 936 (29%; 95% CI: 27–32%) of interactions overall, with 194 of 500 (39%; 95% CI: 35–43%) of interactions at baseline and 81 of 436 (19%; 95% CI: 15–22%) in round 2.

If we further look at our main outcomes by case (Table 3), we see that correct case management was observed in 39 of 250 (16%; 95% CI: 12–21%) of case 1 interactions at baseline and 20 of 218 (9%; 95% CI: 6–14%) in round 2. At baseline, case 2 was correctly managed in 176 of 250 (70%; 95% CI: 64–76%) of interactions and dropped to 96 of 218 (44%; 95% CI: 38–51%) of interactions in the second round of the survey.

Baseline saw ideal case management in 33 of 250 (13%; 95% CI: 10–18%) of case 1 presentations, as compared to 13 of 218 (6%; 95% CI: 4–10%) in round 2. Case 2 was ideally managed in 161 of 250 (64%; 95% CI: 58–70%) of interactions at baseline and only 68 of 218 (31%; 95% CI: 25–38%) in the subsequent round.

Fig 1 details other outcomes of interest, and most notably, medicine prescriptions by case. Case 1 interactions resulted in the dispensation of medicine of any type in 202 of 250 (81%; 95% CI: 75–85%) at baseline and 189 of 218 (87%; 95% CI: 82–91%) of interactions at round 2.

**Table 3. Summary statistics of main outcomes by case.**

| | | Case 1 | | Case 2 | |
|---|---|---|---|---|---|
| | Full sample | Baseline (2014–2015) | Round 2 (2018–2019) | Baseline (2014–2015) | Round 2 (2018–2019) |
| *Correct Case Management* | 0.35 | 0.16 | 0.09 | 0.7 | 0.44 |
| | 331 | 39 | 20 | 176 | 96 |
| | (0.478) | (0.364) | (0.289) | (0.457) | (0.498) |
| *Ideal Case Management* | 0.29 | 0.13 | 0.06 | 0.64 | 0.31 |
| | 275 | 33 | 13 | 161 | 68 |
| | (0.456) | (0.339) | (0.237) | (0.48) | (0.464) |
| *N* | *936* | *250* | *218* | *250* | *218* |

mean coefficients; counts; sd in parentheses

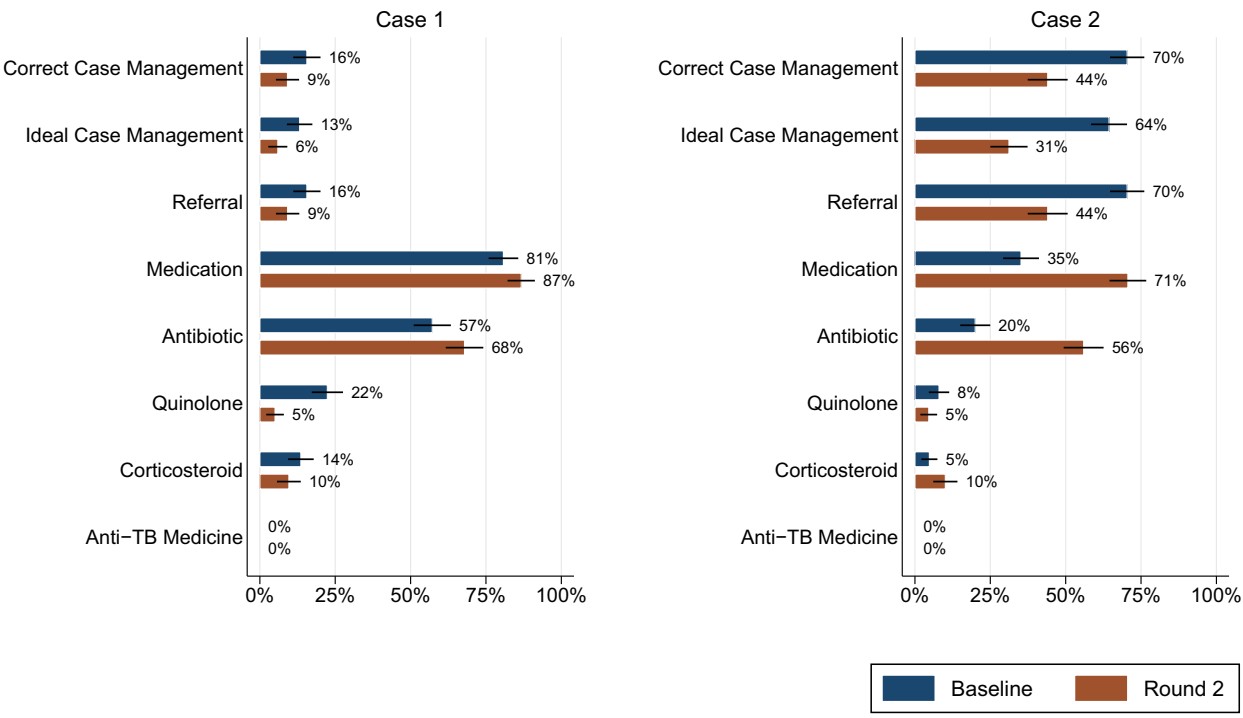

**Fig 1. Proportion of treatment outcomes by case and round of data collection.** All estimates are clustered at the provider level.

A total of 88 of 250 (35%; 95% CI: 30–41%) of case 2 presentations were dispensed medicine at baseline, versus 154 of 218 (71%; 95% CI: 64–76%) in round 2. Dispensation of an antibiotic was seen in 143 of 250 (57%; 95% CI: 51–63%) of case 1 interactions at baseline, and 148 of 218 (68%; 95% CI: 61–74%) at endline. Case 2 SPs were dispensed antibiotics in 50 of 250 (20%; 95% CI: 16–25%) interactions in round 1 and 122 of 218 (56%; 95% CI: 49–62%) of interactions in round 2. A total of 56 of 250 (22%; 95% CI: 18–28%) of case 1 SPs were dispensed quinolone at baseline and 11 of 218 (5%; 95% CI: 3–9%) in the second round. For case 2, quinolones were dispensed in 20 of 250 (8%; 95% CI: 5–12%) of interactions at baseline and 10 of 218 (5%; 95% CI: 3–8%) in the later round. Steroids were given to 34 of 250 (14%; 95% CI: 10–18%) case presentations in round 1 and then to 21 of 218 (10%; 95% CI: 6–14%) of interactions in the next round of data collection. A total of 12 of 250 (5%; 95% CI: 3–8%) of case 2 interactions were given a corticosteroid, in contrast to 22 of 218 (10%; 95% CI: 7–15%) in round 2. Importantly, there were no over-the-counter ATT medicines dispensed by any private pharmacy, across both cases and both rounds.

If we further breakdown components of correct and ideal case management, we can see patterns in referral and dispensation of antibiotic or a corticosteroid. In Fig 2, we see that referrals occurred more often at baseline than in round 2, especially in the management of case 2.

Amongst those who did not refer at baseline, we see that case 1 was managed in the following way: 67 of 250 (27%; 95% CI: 22–33%) of interactions received no antibiotic or corticosteroid; 7 of 250 (3%; 95% CI: 1–6%) received a corticosteroid only; 111 of 250 (44%; 95% CI: 38–51%) received an antibiotic only; and 26 of 250 (10%; 95% CI: 7–15%) received both an antibiotic and corticosteroid.

In round 2, 53 of 217 (24%; 95% CI: 19–31%) did not receive an antibiotic or corticosteroid, 3 of 217 (1%; 95% CI: 0–4%) a corticosteroid only, 125 of 217 (58%; 95% CI: 51–64%) an antibiotic only, and 16 of 217 (7%; 95% CI: 5–12%) an antibiotic and a corticosteroid.

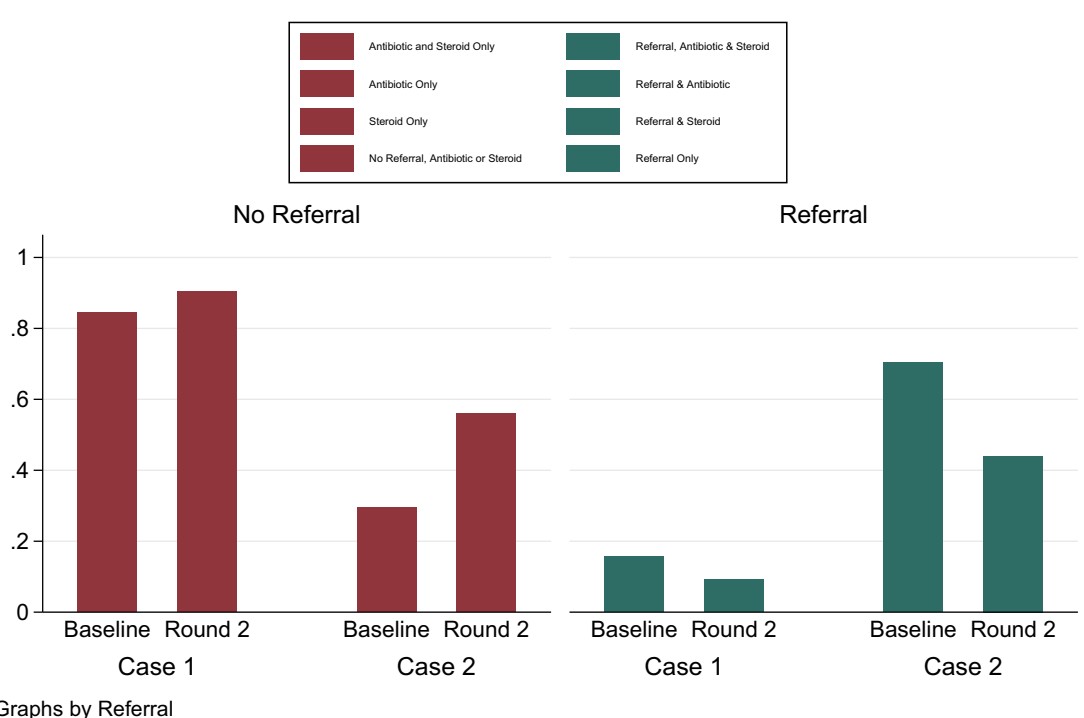

**Fig 2. Changes in referral patterns over round of data collection and dispensation patterns of quinolones and steroids.**

Among pharmacist who were presented with case 2 presentations who did not refer at baseline, 31 of 250 (12%; 95% CI: 9–17%) did not dispense an antibiotic or corticosteroid, 5 of 250 (2%; 95% CI: 1–5%) gave a corticosteroid only, 31 of 250 (12%; 95% CI: 9–17%) an antibiotic only, and 7 of 250 (3%; 95% CI: 1–6%) an antibiotic and corticosteroid only. In round 2 of data collection, among those pharmacist who did not refer, 27 of 218 (12%; 95% CI: 9–17%) interactions additionally did not dispense a corticosteroid or antibiotics, 0 of 218 (0%; 95% CI: 0–2%) a corticosteroid only, 77 of 218 (35%; 95% CI: 29–42%) an antibiotic only, and 18 of 218 (8%; 95% CI: 5–13%) both an antibiotic and a corticosteroid.

Of those providers who referred, case 1 presentations were dealt with in the following ways: at baseline 33 of 250 (13%; 95% CI: 10–18%) solely referred, without the concurrent prescription of an antibiotic or corticosteroid, 0 of 250 (0%; 95% CI: 0–2%) referred and prescribed a corticosteroid only, 5 of 250 (2%; 95% CI: 1–5%) referred and dispensed an antibiotic, and 1 of 250 (0%; 95% CI: 0–2%) provided an antibiotic, corticosteroid and referral. In round 2, for case 1 presentations who were referred, 13 of 217 (6%; 95% CI: 4–10%) did not receive an additional antibiotic or corticosteroid, 0 of 217 (0%; 95% CI: 0–2%) received a corticosteroid only, 5 of 217 (2%; 95% CI: 1–5%) an antibiotic only, and 2 of 217 (1%; 95% CI: 0–3%) an antibiotic, corticosteroid and referral.

For case 2 presentations where providers referred SPs, at baseline 164 of 250 (66%; 95% CI: 60–71%) only referred, 0 of 250 (0%; 95% CI: 0–2%) gave out a corticosteroid, 12 of 250 (5%; 95% CI: 3–8%) an antibiotic and 0 of 250 (0%; 95% CI: 0–2%) a referral, antibiotic and corticosteroid. At endline, 68 of 218 (31%; 95% CI: 25–38%) only referred, 1 of 218 (0%; 95% CI: 0–3%) gave only a corticosteroid, 24 of 218 (11%; 95% CI: 8–16%) only an antibiotic, and 3 of 218 (1%; 95% CI: 0–4%) referred and prescribed an antibiotic and corticosteroid.

We then compared our outcomes of interest for each case by round (Fig 3). We see that for case 1, providers were more likely to both correctly and ideally manage a case at baseline than

in the second round of data collection (OR 0.55; 95% CI: 0.32–0.95 and OR 0.42; 95% CI: 0.22–0.79, respectively). Providers were more likely to prescribe a corticosteroid (OR 0.18; 95% CI: 0.09–0.35) or quinolone (OR 0.67; 95% CI: 0.43–1.06) to SPs as compared to round 2, though the latter was not statistically significant. Providers were however more likely to prescribe any medication or antibiotic in the second round (OR 1.55; 95% CI: 1.04–2.31 and OR 1.58; 95% CI: 1.14–2.20, respectively).

Case 2 witnessed a similar pattern between rounds. Ideal management was more likely to be seen in round 1 of data collection (OR 0.25; 95% CI: 0.17–0.36), as was correct management (OR 0.33; 95% CI: 0.23–0.47). Quinolones were also more likely to be prescribed in the first round of the survey, but this was not statistically significant (OR 0.55; 95% CI: 0.25–1.24). The prescription of any medication (OR 4.43; 95% CI: 3.09–6.35), an antibiotic (OR 5.08; 95% CI: 3.42–7.55) and a corticosteroid (OR 2.23; 95% CI: 1.10–4.50) was more likely to be seen in round 2 than round 1.

We then conducted a difference-in-difference analysis to capture the significant differences in outcomes across the two case presentations, which occur between the baseline and endline periods (Fig 4 and Table 4).

On average, the difference is correct case management between case 1 and case 2 dropped by 20 percent points from baseline to the second round of data collection. Similarly, ideal case management decreased by 26 percentage points. We also see that this is in contrast to the dispensation of medicines, which had the opposite effect between rounds; the difference in dispensation of quinolones between case 1 and case 2 increased by 14 percentage points, as did corticosteroids by 9 percentage points, antibiotics 25 percentage points and medicines generally by 30 percentage points. This highlights the asymmetry in referral rates which constitute

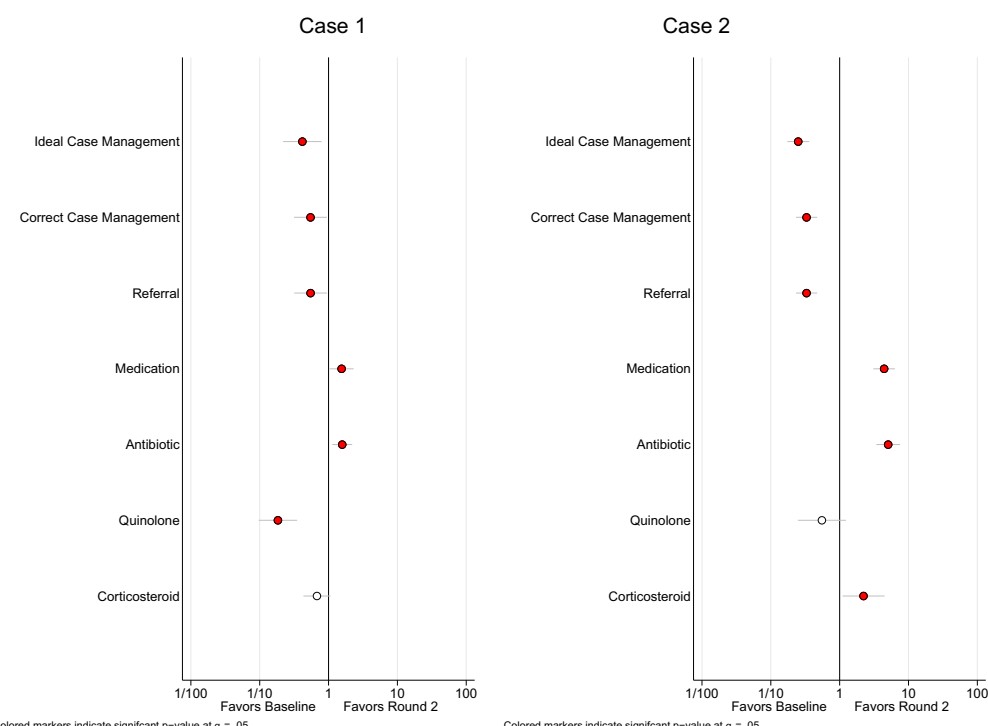

**Fig 3. Outcome changes over time for each respective case; estimates were calculated with an OLS regression, with standard errors clustered at the private pharmacy level.**

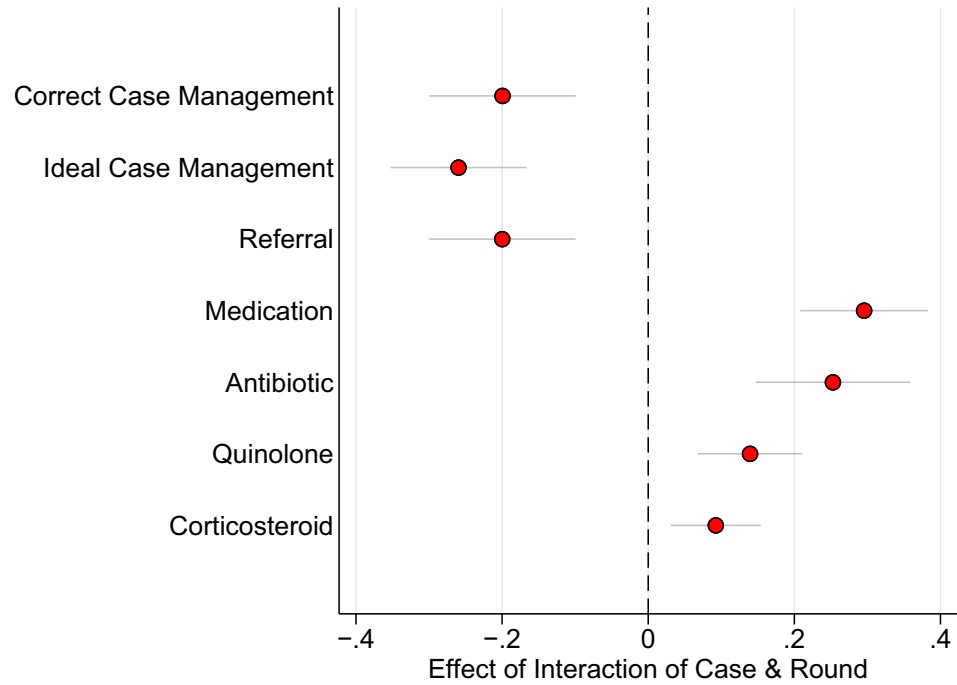

Colored markers indicate signifcant p−value at α = .05.

**Fig 4. Differences-in-difference model by case and round, with standard errors clustered at the provider level.**

the major reason why ideal case management was applied at baseline more than round two, despite the relative impact of prescription of antibiotics and corticosteroids.

We also assessed whether or not providers that demonstrated higher or poorer quality practices were more likely to shut down over time or move. We saw no impact of attrition on the outcome (Fig 5). In addition, we explored the question of whether or not providers who with good quality behaviours in the first round continued to engaged in good quality behaviours in the second round. The effect of consistency of behaviours amongst same providers can be seen in the appendix (Fig 6).

## Discussion

To our knowledge, this is the first study to examine changes in private pharmacy practices over a period of time in a community that had been exposed to efforts directed at greater engagement and training of private health providers. Since we returned after a 5-year period to almost all the same providers initially assessed (some 93%), the findings provide a basis upon which we can judge the merit of current efforts for private sector engagement. Our findings suggested marked improvements as well as disconcerting declines in quality of care, complementing our recent study that assessed tuberculosis management by healthcare providers [17]. On the one hand, we found that no private pharmacy dispensed group 1 anti-tuberculosis treatment medicines. However, when we looked at other metrics of quality, we did not see such comforting findings. Our findings show that ideal case management, where a patient was not only referred but also not dispensed a harmful medication was observed in 33 of 250 (13%; 95% CI: 10–18%) of case 1 presentations at baseline and 13 of 218 (6%; 95% CI: 4–10%) in round 2. Ideal management increased with the addition of more information to the case presentation, with case 2 being ideally managed in 161 of 250 (64%; 95% CI: 58–70%) of

**Table 4. Differences-in-difference model by case and round, with standard errors clustered at the provider level.**

| | CONTROL | | TREATMENT | | DIFFERENCE-IN-DIFFERENCE |
|---|---|---|---|---|---|
| | Baseline | Difference | Baseline | Difference | |
| | Mean | Coef. | Mean | Coef. | Coef. |
| | (SE) | (SE) | (SE) | (SE) | (SE) |
| | Clusters | Clusters | Clusters | Clusters | Clusters |
| **VARIABLE** | N | N | N | N | N |
| **CORRECT CASE MANAGEMENT** | 0.16 | -0.06** | 0.7 | -0.26*** | -0.20*** |
| | (-0.02) | (-0.03( | (-0.03) | (-0.04) | (-0.05) |
| | 250 | 258 | 250 | 250 | 260 |
| | 250 | 468 | 250 | 468 | 936 |
| **IDEAL CASE MANAGEMENT** | 0.13 | -0.07*** | 0.64 | -0.33*** | -0.26*** |
| | (-0.02) | (-0.03) | (-0.03) | (-0.04) | (-0.05) |
| | 250 | 258 | 250 | 250 | 260 |
| | 250 | 468 | 250 | 468 | 936 |
| **REFERRAL** | 0.16 | -0.06** | 0.7 | -0.26*** | -0.20*** |
| | (-0.02) | (-0.03) | (-0.03) | (-0.04) | (-0.05) |
| | 250 | 258 | 250 | 250 | 260 |
| | 250 | 467 | 250 | 468 | 935 |
| **MEDICATION** | 0.81 | 0.06** | 0.35 | 0.35*** | 0.30*** |
| | (-0.02) | (-0.03) | (-0.03) | (-0.04) | (-0.04) |
| | 250 | 258 | 250 | 250 | 260 |
| | 250 | 468 | 250 | 468 | 936 |
| **ANTIBIOTICS** | 0.57 | 0.11*** | 0.2 | 0.36*** | 0.25*** |
| | (-0.03) | (-0.04) | (-0.03) | (-0.04) | (-0.05) |
| | 250 | 258 | 250 | 250 | 260 |
| | 250 | 468 | 250 | 468 | 936 |
| **QUINILONE** | 0.22 | -0.17*** | 0.08 | -0.03 | 0.14*** |
| | (-0.03) | (-0.03) | (-0.02) | (-0.02) | (-0.04) |
| | 250 | 258 | 250 | 250 | 260 |
| | 250 | 468 | 250 | 468 | 936 |
| **CORTICOSTEROID** | 0.14 | -0.04* | 0.05 | 0.05** | 0.09*** |
| | (-0.02) | (-0.02) | (-0.01) | (-0.02) | (-0.03) |
| | 250 | 258 | 250 | 250 | 260 |
| | 250 | 468 | 250 | 468 | 936 |

interactions at baseline and 68 of 218 (31%; 95% CI: 25–38%) in the subsequent round. Since standardised patient methodology systematizes the presentation of the underlying clinical presentation across different providers [27], the results can be considered reliable, valid, and comparable across private pharmacies.

A key repeat finding is that none of the private pharmacies in our study dispensed group 1 anti-tuberculosis drugs without a prescription. Fortunately, any concerns regarding the use of anti-tuberculosis drugs by private pharmacies seem to be unfounded and reports that private pharmacies are sources for TB resistance via the inappropriate dispensation of anti-tuberculosis are unlikely. While we did not have the opportunity to conduct qualitative research on private pharmacy decision-making, it is possible that TB drugs, unlike antibiotics and corticosteroids, are considered toxic, and that tuberculosis requiring long-term treatment might play a part in private pharmacies being cautious about OTC use of ATT drugs. Findings are in stark contrast to the results found in a recent publication by our group in private clinical

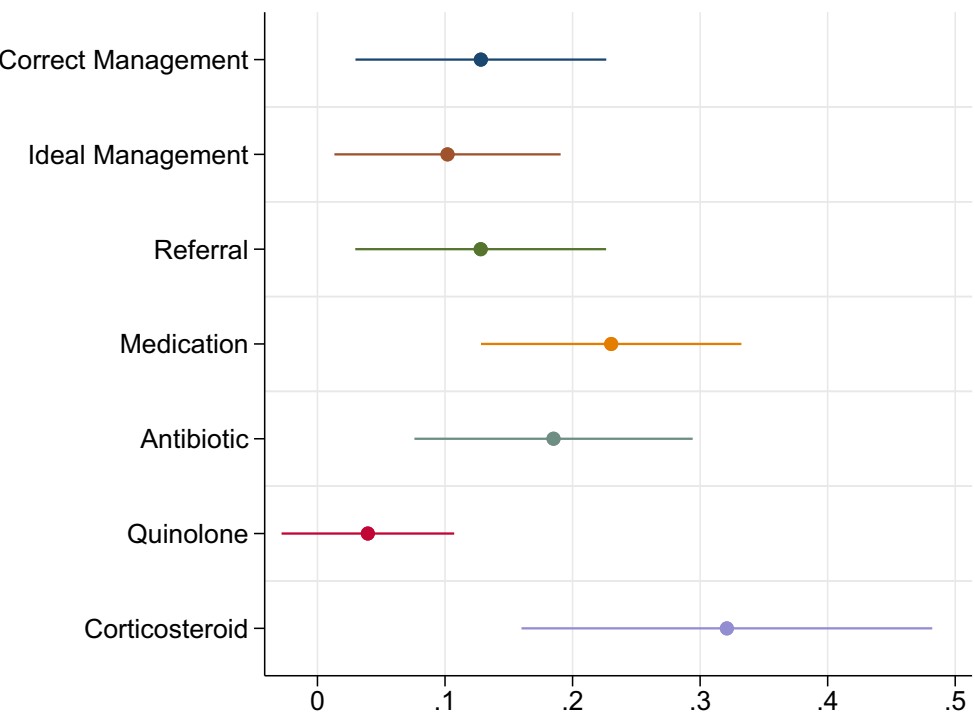

**Fig 5. Consistency of behaviours amongst same providers, calculated using a linear regression model, with standard errors clustered at the provider level.**

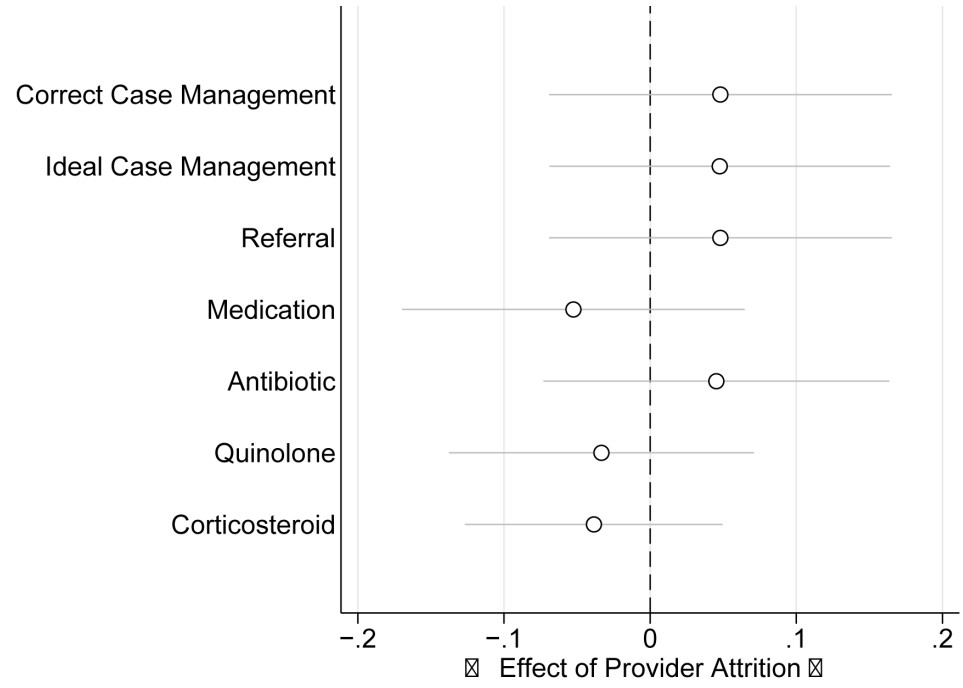

Colored markers indicate signifcant p−value at α = .05.

**Fig 6. Impact of attrition on outcome calculated using a linear regression model, with standard errors clustered at the provider level.**

care facilities in Patna and Mumbai which found that one in five SPs presenting with abnormal CXR were prescribed ATT by private providers [30]. Perhaps it may also be in part due to the proactiveness of the Indian National Tuberculosis Control Program in including tuberculosis drugs under Schedule H-1, subsequently requiring providers to maintain a register and report these prescriptions [38].

Our results indicate a decline in progress between rounds of data collection by case. If we define outcomes as successful based on referral, we see that the propensity to refer patients–the primary metric used to evaluate correct treatment in each study–faced a decline from baseline. This decline in referral rates with concurrent upsurge prescription of potentially harmful medicines is disconcerting; the increased use of quinolones and corticosteroids is especially worrying because these drugs delay tuberculosis diagnosis [39, 40]. Miller et al. reviewed the literature on SP studies among private pharmacies for TB, and showed similar poor quality of care across various settings [23].

Confirmed diagnoses dictate what pharmacists do, and sharp increases in ideal management and large decreases between case 1 and case 2 in antibiotic use are indicative of that. This large difference suggests that the main challenge faced by pharmacists is confusion about the likely diagnosis. Better training regarding tuberculosis symptoms and encouraging early referrals for patient with tuberculosis symptoms might help. The TB PPM Learning Network has worked on a simple infographic tool that could be adapted by countries and used to educate private pharmacies on the dos and don'ts while dealing with people with TB cough as the primary complaint (Fig 5).

Even as rates of ideal management went down over time, management of case 2 (where a diagnosis was confirmed) over case 1 (which relied on symptoms alone) were higher by 5-fold at both time points. This suggests best practices can be supported by a confirmed diagnosis. But, given that a high proportion of SPs were poorly managed even in light of a confirmed diagnosis, significantly greater inquiry into private pharmacy practices and interventions that address incorrect practices are needed. This warrants much stronger engagement with the private pharmacy sector, who, in light of broader PPM initiatives, may be neglected. This underscores the fact that the use of antibiotics is mediated by drug category and the information that patients present, something that was observed in our first study, and while this effect was tapered in subsequent round, it still holds true.

Quinolones are also an integral part of multidrug-resistant tuberculosis treatment regimens and emerging regimens, so such suboptimal OTC use of this class of medications are of concern [40]. The widespread use of antibiotics and corticosteroids for respiratory symptoms also has implications for community-acquired infections more generally. Inappropriate use of quinolones is a major risk factor for producing highly resistant Gram-negative enteric bacteria, resulting in an increased risk of diarrhoeal illness, bacteraemia, and other infections, especially in India [41]. Private pharmacies often dispense antibiotics for clients with chronic cough without referral for TB testing, with the expectation of improving client retention [14, 18]. This problem could possibly be addressed by supplying private pharmacies with referral slips, which could allow private pharmacies to dispense a product while maintaining their position relative to client expectations of action [42]. The use of broad spectrum antibiotics for respiratory symptoms identified in our study might contribute to resistant strains of common respiratory pathogens [43]. Unnecessary use of corticosteroids is associated with an increased risk of developing lower respiratory tract infection, cellulitis, herpes zoster, and candidiasis, in addition to potentially delaying treatment [44].

In countries with a high burden of tuberculosis, private pharmacies are frequently the first point of care seeking for individuals with TB indicative symptoms [13, 15, 16, 45, 46]. This holds true in India. India has a high rate of drug-resistant pathogens, likely driven by rampant

antibiotic misuse, a high burden of infectious diseases, easy access to antibiotics, and a fragmented, unregulated, privatized healthcare system. While all antibiotics, anti TB medications, and corticosteroids in India are listed in Schedule H under the Ministry of Health and Family Welfare Department of Health's Drugs and Cosmetics Rules, 1945, and dispensing them requires a valid prescription, due to weak enforcement, pharmacists readily give antibiotics to patients without prescriptions, as shown in our previous SP surveys of private pharmacies in India [18, 47]. The quality of TB care by private pharmacies in low-income and middle-income countries (LMICs) has been shown to be low in other contexts, with lack of TB knowledge among private pharmacy staff, inappropriate sales of antibiotics and anti-TB medications, and lack of systems to facilitate referrals for TB testing [14, 18, 23]. Most private pharmacies are not linked to national tuberculosis programmes (NTPs) but establishing structured mechanisms for the referral of individuals with presumptive TB, not just the mandatory notification of ATT purchases, from private pharmacies to private or NTP-associated facilities for testing could help to identify the millions of missing patient who develop TB yearly but were not diagnosed and reported to NTPs [48].

Since it was beyond the scope of our study to conduct qualitative research, it is unknown why some pharmacists give antibiotics. Moreover, it is unclear whether the variation in our data is explained by the qualification of the person providing advice in private pharmacies. A typical private pharmacy in India has a qualified pharmacist, and many other helpers and assistants who are not qualified in private pharmacy science. It is unclear what advice pharmacists provide vis-à-vis their helpers, since SP studies are unable to discern the exact qualifications of the person interacting with the standardized patients. There are some clues within the qualitative literature, suggesting that a combination of other factors might also be at play, including pharmaceutical industry marketing techniques, business models followed by local providers, and active demand from patients for medicines [49]. It has been shown that the reciprocal relationships between the retail private pharmacists, private providers, medical representatives and the prevalence of kickbacks (financial incentives) influenced pharmacist drug-stocking patterns [50]. Overstock, near-expiry, and undersupply are additional factors which may explain the misuse of antibiotics and restricted drugs [47, 49].

Engaging with the private sector in TB management is important; encouraging NTPs to develop strong public-private partnerships is highlighted by the WHO as a top priority in the End TB strategy [51]. Given that 50% of India's TB is managed outside of the public sector [52, 53], engagement with private providers is essential. The Revised National TB Control Programme (RNTCP) has implemented public-private mix (PPM) programmes through private-provider interface agencies (PPIAs) [54] These initiatives could extend to the private pharmacy system as well, since, despite engagement of the private sector, our work shows that those efforts may be missing the mark. Additional work is needed to uncover why this is happening, that provider practise may not be responding to the incentives and trainings being doted out, meaning a better evaluation and revisiting of those initiatives may be needed. Robust and effective private sector engagement is ultimately predicated on behavioural change on the part of providers, which may not be as simply resolved through trainings alone. Lessons learnt from an evaluation of private pharmacy interventions have shown that to properly retain pharmacists in such initiatives over time, a package of incentives was required that included not only monetary incentives but also training in TB and the opportunity to interact with influential stakeholders in ways which were seen by private pharmacy owners as beneficial to their businesses. This is particularly true due to the fact that there is pressure from clients themselves; clients expect to receive a tangible outcome such as medication from a visit to a private pharmacy, even in the absence of diagnostic testing [42]. Creating sustainable, long term linkages between private pharmacies and referral facilities are vital success, and information

should flow in both directions [20]. This would be facilitated by digital tracking tools for private pharmacies to facilitate information sharing and real-time updates and received feedback on the number of their referred clients attending testing sites and the number diagnosed with TB [42].

Our study is not without limitations. First, our repeat study could only be conducted in one city, Patna, so our findings may not be generalizable to other cities in India and our study does not provide evidence on how pharmacists in rural areas manage patients with tuberculosis or tuberculosis symptoms. Second, our study reflects what happens when pharmacists receive a completely unknown patient as opposed to a known, regular client, or a client who returns to the pharmacist after one round of ineffective treatment, as would be expected from many community private pharmacies; such familiarity could well have mediated some of the behaviours. Third, differences between case 1 and case 2 outcomes could reflect variation in the standardised patient profile themselves; we could not assess this possibility given that different standardised patients were assigned to the two cases with no crossover. In other studies however, it has been shown that inclusion of standardised patient characteristics has little effect on estimated coefficients and coefficients remain stable when we account for standardised patient sex, height, and weight [18]. Also, our SP studies could not identify whether the interactions were done with qualified pharmacists or their helpers/assistants. It is likely that most interactions were with private pharmacy staff rather than qualified pharmacists.

## Conclusion

Our standardised patient study provides valuable insights into how private pharmacies in an urban Northern Indian city changed their treatment of patients with tuberculosis symptoms or with confirmed tuberculosis over a 5-year period. We saw that overall, private pharmacy performance weakened over time. Importantly, however, no over-the-counter dispensation of anti-TB medications occurred in either survey round. As the first point of contact for many care seekers, continued and sustained efforts to engage with Indian private pharmacies should be prioritized.

## Author Contributions

**Conceptualization:** Anita Svadzian, Jishnu Das, Veena Das, Ranendra Das, Madhukar Pai.

**Data curation:** Anita Svadzian, Benjamin Daniels, Giorgia Sulis, Ada Kwan.

**Formal analysis:** Anita Svadzian, Benjamin Daniels, Giorgia Sulis.

**Funding acquisition:** Jishnu Das, Madhukar Pai.

**Methodology:** Jishnu Das, Amrita Daftary, Ada Kwan, Ranendra Das, Madhukar Pai.

**Project administration:** Ranendra Das.

**Supervision:** Jishnu Das, Amrita Daftary, Veena Das, Madhukar Pai.

**Writing – original draft:** Anita Svadzian.

**Writing – review & editing:** Benjamin Daniels, Giorgia Sulis, Jishnu Das, Amrita Daftary, Ada Kwan, Veena Das, Ranendra Das, Madhukar Pai.

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
