## [Decision Letter · Decision Letter 0]

22 Mar 2023

PGPH-D-23-00255

Use of standardised patients to assess tuberculosis case management by pharmacies in Patna, India: a repeat cross-sectional study

Dear Dr. Pai,

Thank you for submitting your manuscript to PLOS Global Public Health. After careful consideration, we feel that it has merit but does not fully meet PLOS Global Public Health’s publication criteria as it currently stands. Therefore, we invite you to submit a revised version of the manuscript that addresses the points raised during the review process.

We look forward to receiving your revised manuscript.

Kind regards,

Dan Kajungu, PhD

Academic Editor

Journal Requirements:

Additional Editor Comments (if provided):

Reviewers' comments:

Reviewer's Responses to Questions

**Comments to the Author**

1. Does this manuscript meet PLOS Global Public Health’s publication criteria? Is the manuscript technically sound, and do the data support the conclusions? The manuscript must describe methodologically and ethically rigorous research with conclusions that are appropriately drawn based on the data presented.

Reviewer #1: Yes

Reviewer #2: Yes

2. Has the statistical analysis been performed appropriately and rigorously?

Reviewer #1: Yes

Reviewer #2: Yes

3. Have the authors made all data underlying the findings in their manuscript fully available (please refer to the Data Availability Statement at the start of the manuscript PDF file)?

Reviewer #1: Yes

Reviewer #2: Yes

4. Is the manuscript presented in an intelligible fashion and written in standard English?

Reviewer #1: Yes

Reviewer #2: Yes

5. Review Comments to the Author

Reviewer #1: Review: Use of standardised patients to assess tuberculosis case management by pharmacies in Patna, India: a repeat cross-sectional study.

Overall

The manuscript is well written, technically sound and covers an important topic.

In my view the statistical analyses are appropriate and rigorous but further review by a statistician might be helpful.

The authors report that the datasets are available and there is a link in the manuscript where you can request for study tools. It is not clear to me how you obtain the datasets.

Comments to the authors

With likely more 70% similarity to the preceding study, shouldn’t this manuscript have been published as update to the previous one?

Reading the manuscript, the information you want is all there but not where you would expect it for example:

1. Some key information like “what a standardised patient is” should be provided in the introduction as part of the summary of the previous study. As it is now, it leaves you guessing until much further in the manuscript.

2. Paragraph 463-469 should come after 437-446 then 513-524

A bit more distinction between the 2 studies is needed maybe in a separate paragraph or table. The authors mentioned that 92% of the same pharmacies were used; were the SOPs, standardized patients, and study teams the same? If not what were the differences and what potential impact could these have had?

Lines 140-144… consider integrating this into the introduction section.

Lines 212 to 214, how many standardized patients were used in the second round? This doesn’t come out clearly.

Line 217 PPIA should be defined here and not 224-225.

Figures: Please add headings and label all the axes.

The manuscript should be shortened e.g. by presenting only the key results and others left to an appendix.

In lines 526-545 the authors prescribe generic solutions and yet the study did not investigate the reasons for the decline in ideal and correct case management. This section should be removed or shortened because it is speculative.

Reviewer #2: Comments to the authors

Abstract and Introduction

Some sentences in the abstract are too long and confusing. The authors need to revise and cut short sentences for easy flow of the abstract: consider lines 44 to 48.

The first and second sentences in the methods abstract section (lines 51-55) are more of study objectives, this can be pushed to the end of the introduction.

The authors leave the reader hanging in their methods section of the abstract. A lot is missed out, issues on when the two surveys were conducted, sampling for the selected pharmacies, ought to come out clearly.

Generally, the introduction is well written, flows well and illustrates the gap and study problem.

The terms “private pharmacies” and “pharmacies” are interchangeably used throughout the paper. The authors need to use one term for consistence, remember public pharmacies also exist. For example, consider introduction line 99 to 105.

What does the term “interaction” mean in the paper? Is this patient-pharmacist interaction? It is not very clear and leaves the reader wondering at first read.

Is it okay to use the term standardized participant for patients presenting with classic symptoms of pulmonary TB? Is this a standard thing? It is not clear who a standardized patient is at first read in the introduction

Line 86, there is a repetition of the word “In”, revise this

Line 86 to 89, there is poor consistence and coherence between the sentences, consider revising the joining sentence of “These people, considered ‘missing’, are either undiagnosed or not notified to the TB programme, and comprise around 4.2 million cases…”. One wonders where these people come from; feels like they are already talked about yet they are not yet.

Methods

The methods are well detailed and explained fully.

Now the term “interactions” clearly comes out in the methods. It would be better to rephrase it to patient-pharmacist interaction the first time it appears in the paper for the reader to easily follow.

Table 1 can be pushed to supplementary materials

Line 213, it is not clear how the authors come up with the fixed sample size of 805 additional interactions.

Definition of the outcome: lines 254-258: How well did you obtain information regarding the verbal referral, what was the proportion of those who had received the verbal referral; is this reliable anyway?

Statistical analysis: Line 261-262; The sentence is in future tense, yet this was already done, please revise it.

Results

Results are well presented and explained; good work done.

Discussion and conclusion

Line 437 to 448; second paragraph is doing a lot of describing results which was already done in the results section; authors need to summarize the results here and compare and contrast their results with previous studies in similar contexts and give implications for their results.

Line 463 to 469; While the authors mention the decline in progress between rounds of data collection by case and give it’s implications, they do not explain the possible explanation for the decline amidst the interventions done in the country. Adding a possible explanation adds value to this.

General comment: This is a good paper that adds value to the TB knowledge and pharmacy prescription practices.

6. PLOS authors have the option to publish the peer review history of their article (what does this mean?). If published, this will include your full peer review and any attached files.

**Do you want your identity to be public for this peer review?** For information about this choice, including consent withdrawal, please see our Privacy Policy.

Reviewer #1: No

Reviewer #2: No

---

## [Editor Report · Decision Letter 1]

21 Apr 2023

Use of standardised patients to assess tuberculosis case management by private pharmacies in Patna, India: a repeat cross-sectional study

PGPH-D-23-00255R1

Dear Dr. Pai,

We are pleased to inform you that your manuscript 'Use of standardised patients to assess tuberculosis case management by private pharmacies in Patna, India: a repeat cross-sectional study' has been provisionally accepted for publication in PLOS Global Public Health.

Best regards,

Dan Kajungu, PhD

Academic Editor